# Multivariate Sparse Coding of Nonstationary Covariances with Gaussian Processes

Rui Li

Golisano College of Computing and Information Sciences
Rochester Institute of Technology
Rochester, NY 14623
`rxlics@rit.edu`

## Abstract

This paper studies statistical characteristics of multivariate observations with irregular changes in their covariance structures across input space. We propose a unified nonstationary modeling framework to jointly encode the observation correlations to generate a piece-wise representation with a hyper-level Gaussian process (GP) governing the overall contour of the pieces. In particular, we couple the encoding process with automatic relevance determination (ARD) to promote sparsity to account for the inherent redundancy. The hyper GP enables us to share statistical strength among the observation variables over a collection of GPs defined within the observation pieces to characterize the variables' respective local smoothness. Experiments conducted across domains show superior performances over the state-of-the-art methods.

## 1   Introduction

In many real-world applications, multivariate observations exhibit critical irregular changes in their covariance smoothness with sharp transitions. For example, a major challenge to accurately locate a seizure-onset zone (SOZ) through intracranial electroencephalography (iEEG) recordings is to detect different forms of sudden transient electrophysiologic events of SOZ signals [1, 2, 3]. Another scenario is that regional outbursts and geographic features (e.g., parks, rivers) lead to complex tempo-spatial variations of crime occurrences across regions over time [4, 5].

In these scenarios, some segments of the observations exhibit larger variability than others. The stationary methods, assuming the same covariance structure throughout the entire input space, cannot capture such change in covariance smoothness. Conventional nonstationary modeling methods are limited to model univariate observation by two consecutive steps [6, 7, 8]. They first recursively partition the input space into regions, and then define separate local Gaussian processes (GPs) within each region. The GP inference step cannot capture long-range dependence or share statistical information among the independent local GPs. A solution to alleviate the problem is to combine the local GPs with a global GP which is fitted to the whole observation [9, 10]. However, it tends to over-smooth the local covariance variability.

To address these challenges, we propose a novel nonstationary modeling framework that jointly infers a piece-wise representation of the multivariate observations and a hyper-level GP governing the overall contour of the pieces. In particular, we employ multilogit regression function to encode the observation correlations coupled with automatic relevance determination (ARD) priors over the coefficients to promote sparsity. This encoding process transforms the observations into a set of disjoint pieces to model the variability in the covariance smoothness with the correlations providing between-variable information. Since commonly only a portion of observations are informative for such transformation, ARD shrinks the correlation dimensions towards zero to handle the inherent

redundancy. Regulated by the hyper GP through their mean functions, a collection of variable-specific local GPs are defined to model the variables' respective smoothness within the pieces. The hyper GP not only shares statistical strength across the local GPs while retaining their distinctive covariance property, but also induces the observation variables' conditional independence. The piece-wise representation leads to efficient posterior computation with the conjugate priors.

We evaluate our nonstationary modeling method across domains: for seizure onset localization, we achieve robustly better performances than the state-of-the-art competing methods; for crime occurrence prediction, by modeling the evolving covariances of weekly crime rates among the 179 census tracts in Washington D.C between 2015-2019, we outperform the state-of-the-art methods.

## 2  Related work

Although iEEG recordings provide critical information to locate areas of the brain to remove for epilepsy patients, pre-surgical examination of between-seizure iEEG signals is a labor-intensive and error-prone process [1, 11]. It becomes increasingly essential to develop effective computational methods to identify the iEEG channels that are most likely to be in the SOZs by identifying different abrupt changes in neurophysiological events [2, 3, 12, 13]. Empirical studies focus on identifying biomarkers (e.g., spectral features, high frequency oscillations (HFO)) related to sub-clinical epileptic bursts [1, 12]. Classical modeling methods make stationary assumption without considering covariance change over time. In [13], Markov switching process is coupled with a stochastic process prior to analyze the iEEG signal dynamics. A factor graphical model is proposed to integrate temporal and spatial information of iEEG channels to infer pathologic brain activity for SOZ localization [3]. Specifically, the spatial property is defined as correlations between channels, and the temporal function measures correlations between a channel's current state and the linear combination of its previous states. GP with stationary covariance is applied to model nonlinearity in neonatal EEG signals for seizure detection, and shows high level of prediction performance [14]. For crime prediction, an autoregressive mixture model with Poisson processes is proposed [5]. Its most recent extension PoINAR incorporates a stochastic process prior to group spatial correlation modes across multiple time series, and achieves the state-of-the-art performance [15].

Nonstationary covariance function modeling methods with designed or learned kernels typically assume the same covariance structure as a function of distances from observations throughout the input space [16, 17]. This is a strong modeling assumption for the above applications where sharp transitions in covariance smoothness play the key role. Partitioning including Bayesian trees, Voronoi tessellation, and normalized cuts (N-cuts) is widely used for modeling nonstationarity with abrupt changes [6, 7, 8, 9]. The local GPs defined within the recursively partitioned regions are independent. To capture the long-range trend, some methods define a global stationary GP over the entire input space, and combine it with the local GPs. This leads to over-smooth the complex covariances induced by the local GPs, since the global GP is also independent of the partition inference procedure [9, 10]. Additionally, these methods are subject to some constraints such as partition points having to be at observation locations, and balanced binary trees. A mixture of GP experts models nonstationary univariate observations by defining each GP expert over the entire input space [18, 19].

## 3  Unified nonstationary modeling framework

Our framework encodes observation correlations into a trending piece-wise representation with both ARD and hyper GP priors. By coupling the relevance vectors with the hyper GP, we are able to share statistical information among the pieces. Given the hyper GP, each observation variable is modeled by a conditionally independent GP within the pieces for its local covariance smoothness.

### 3.1  Sparse coding for observation correlations

Let $Y = \{y_1, \cdots, y_N\}$ denote a set of multivariate observations at locations $\{x_1, \cdots, x_N\}$ with $x_i \in X$ as a non-random covariate in the input space $X$ and an observation $y_i \in R^{D \times 1}$. We encode $Y$ into $K$ pieces with the corresponding inputs $X = \cup_k X_k$ and $X_k \cap X_{k'} = \emptyset$, where $k \neq k'$.

Let $Z$ denote a $N \times K$ indicator matrix. Its element $z_{ik} = \delta(x_i \in X_k)$ is the one-of-$K$ encoding of $x_i$, where $z_{ik}$ turns on iff $x_i \in X_k$. Its probability of being 1 is a multilogit regression function:

$$p(z_{ik} = 1|\theta_k, Q) = \frac{\exp(\theta_k^T Q(\cdot, y_i))}{\sum_{k'=1}^K \exp(\theta_{k'}^T Q(\cdot, y_i))} \tag{1}$$

where $\theta_k$ denotes a $N \times 1$ coefficient vector for the $k$'th piece, and $Q(\cdot, y_i)$ is the $i$'th column vector of the observation correlation matrix $Q$.

We employ the sparse prior ARD to explore how the correlation between any two observations contributes to the encoding. ARD eliminates the irrelevant correlations by encouraging their coefficients go to zero. Specifically, we define independent, zero-mean, spherically symmetric Gaussian priors on $\theta_k$:

$$p(\theta_k|\alpha_k) = N(\theta_k|0, A_k^{-1}) \tag{2}$$

where $A_k^{-1} = diag(\alpha_k^{-1})$ denotes a diagonal matrix with the components of vector $\alpha_k^{-1}$ on the diagonal. Each component of precision parameter $\alpha_k$ is given a $\Gamma(a, b)$ prior. ARD method penalizes non-zero coefficient components by an amount determined by the precision parameters. Iterative estimation of $\alpha_k$ and $\theta_k$ leads to $\alpha_k$ becoming large for components whose evidence in the correlations is insufficient for overcoming the penalty induced by the prior. Having $\alpha_k \to \infty$ drives $\theta_k \to 0$, which implies that the corresponding correlations do not contribute to the encoding. Therefore, ARD identifies a subset of the observations, known as relevance vectors, with non-zero coefficients for each piece.

Let $v_k$ denote the input in $X_k$ whose corresponding observation is the relevance vector with the maximum absolute value of non-zero component of $\theta_k$, where $v_k \in V \subset X$. We define a function: $g : V \to \mathcal{R}$ which describes the overall contour of the observation pieces by sharing statistical information among them:

$$g(v) \sim GP(0, \kappa_g(v, v')) \tag{3}$$

where $\kappa_g$ is a covariance function defined on $V$. We use a squared-exponential kernel $\kappa_g = \sigma_g^2 \exp(-l_g\|v - v'\|_2^2)$ to encourage a smooth profile of the pieces. We further define a local function $f_k : X_k \to \mathcal{R}$ for each piece:

$$f_k(x)|g \sim GP(g(v_k), \kappa_k(x, x')) \tag{4}$$

where $g(v_k)$ specifies the mean function of the GP prior for the local function $f_k$. $\kappa_k$ is a squared-exponential kernel $\kappa_k = \sigma_k^2 \exp(-l_k\|x - x'\|_2^2)$ defining a covariance function. We assume $l_k = \frac{l_g}{\|X_k\|_2^2}$ to let the horizontal lengthscales of the local functions reflect the global smoothness.

## 3.2 Piece-wise GPs for univariate observations

Let $\mathbf{g} = g(V) \in R^{K \times 1}$ and $\mathbf{f} = [f_1(X_1)^T, \cdots, f_K(X_K)^T]^T \in R^{N \times 1}$, the hyper-level and local GPs define two joint Gaussians for any finite set of observations, respectively:

$$p(\mathbf{g}|V) = N(\mathbf{g}|0, \Sigma_g) \qquad p(\mathbf{f}|\mathbf{g}, Z) = N(\mathbf{f}|Z\mathbf{g}, \Sigma_f) \tag{5}$$

where $\Sigma_g$ is the covariance matrix with $\kappa_g(v, v')$ as the elements, and $\Sigma_f$ is a diagonal block covariance matrix in which the elements of the $k$'th block $\Sigma_f^{(k)}$ are $\kappa_k$ of the input pairs in the $k$'th piece as $[\Sigma_f^{(k)}]_{ij} = \kappa_k(x_i^{(k)}, x_j^{(k)})$, where $x_i^{(k)}, x_j^{(k)} \in X_k$.

One can analytically marginalize $\mathbf{g}$ conditioned on the piece-wise representation $Z$ yielding

$$p(\mathbf{f}|Z) = N(\mathbf{f}|0, Z\Sigma_g Z^T + \Sigma_f) \tag{6}$$

A univariate observation $y \in R^{N \times 1}$ with noise is thus generated as:

$$p(y|\mathbf{f}, \sigma^2) = N(y|\mathbf{f}, \sigma^2 I) \tag{7}$$

where $I$ is a $N \times N$ identity matrix. Recalling (6), the marginal likelihood conditioned on $Z$ yields

$$p(y|Z) = \int p(y|\mathbf{f})p(\mathbf{f}|Z)d\mathbf{f} = N(y|0, \Sigma_y) \tag{8}$$

where $\Sigma_y = Z\Sigma_g Z^T + \Sigma_f + \sigma^2 I$ denotes the induced nonstationary covariance matrix. $\Sigma_y$ captures the varying covariance structures of the pieces, and the discontinuities between them.

## 3.3 Piece-wise GPs for multivariate observations

To extend to multivariate observations $Y = \{y^{(1)}, \cdots, y^{(D)}\}$ where $y^{(d)} \in R^{N \times 1}$ denotes the observations of the $d$th variable (e.g., a feature of iEEG recording, a census tract's crime occurrences), we model each variable's observations $y^{(d)}$ as a realization from a specific local function $\mathbf{f}^{(d)}$ as

$$p(y^{(d)}|\mathbf{f}^{(d)}, \sigma^2) = N(y^{(d)}|\mathbf{f}^{(d)}, \sigma^2 I) \tag{9}$$

The variable-specific local functions $\{\mathbf{f}^{(d)}\}$ are conditionally independent given $\mathbf{g}$ and $Z$:

$$
\begin{aligned}
p(\mathbf{f}^{(1:D)}|\mathbf{g}, Z) &= \prod_{d=1}^{D} p(\mathbf{f}^{(d)}|\mathbf{g}, Z) \\
&= \frac{|2\pi\frac{\Sigma_f}{D}|^{\frac{1}{2}}}{|2\pi\Sigma_f|^{\frac{D}{2}}} \exp[-\frac{1}{2}tr(\sum_d (\mathbf{f}^{(d)} - \bar{\mathbf{f}})^T \Sigma_f^{-1}(\mathbf{f}^{(d)} - \bar{\mathbf{f}}))] N(\bar{\mathbf{f}}|Z\mathbf{g}, \frac{\Sigma_f}{D})
\end{aligned}
\tag{10}
$$

where $\bar{\mathbf{f}} = \frac{\sum_d \mathbf{f}^{(d)}}{D}$. The assumption allows to share statistical strength among the observation variables through $\mathbf{g}$ while retaining variable-specific covariance variability, as each variable's observations can be derived by marginalizing over $\mathbf{f}^{(d)}$:

$$p(y^{(d)}|\mathbf{g}, Z, \sigma^2) = \int p(y^{(d)}|\mathbf{f}^{(d)}, \sigma^2 I)p(\mathbf{f}^{(d)}|\mathbf{g}, Z)d\mathbf{f}^{(d)} = N(y^{(d)}|Z\mathbf{g}, \Sigma_{y|g}) \tag{11}$$

where $\Sigma_{y|g} = \Sigma_f + \sigma^2 I$. By exploiting the conditional independence of $Y$, the marginal likelihood for the multivariate observations is:

$$
\begin{aligned}
p(Y|Z, \sigma^2) &= \int \prod_{d=1}^{D} p(y^{(d)}|\mathbf{g}, Z, \sigma^2)p(\mathbf{g}|V)d\mathbf{g} \\
&= \frac{|2\pi\frac{\Sigma_{y|g}}{D}|^{\frac{1}{2}}}{|2\pi\Sigma_{y|g}|^{\frac{D}{2}}} \exp[-\frac{1}{2}tr(\sum_d (y^{(d)} - \bar{y})(y^{(d)} - \bar{y})^T)\Sigma_{y|g}^{-1}] N(\bar{y}|0, \frac{\Sigma_{y|g}}{D} + Z\Sigma_g Z^T)
\end{aligned}
\tag{12}
$$

where $\bar{y} = \frac{\sum_d y^{(d)}}{D}$. The $k$th diagonal block of the covariance matrix is $[\Sigma^{(k)}]_{ij} = \kappa_g(v_k, v_k) + [\kappa_k(x_i^{(k)}, x_j^{(k)}) + \sigma^2\delta(i,j)]/D$. The multivariate case in (12) can be reduced to (8) when $D = 1$. The computation complexity for (12) is $O(KM^3)$, where $M$ denotes the rough size of each piece. By optimizing (12), we can determine the settings of the hyperparameters $\{l_g, \sigma_g^2, \sigma_f^2\}$ [1].

## 3.4 Efficient inference

We develop a Gibbs sampling solution to iteratively sample the GP functions and the piece-wise representation given their priors and the observations, and then update the hyper-parameters given the latent functions and the observations.

First, our model's joint probability can be factorized as

$$
\begin{aligned}
&p(Y, \{\mathbf{f}^{(1:D)}\}, \mathbf{g}, Z, \{\theta_k\}, V, X, \sigma^2, Q, \alpha) \\
&\propto \prod_{d=1}^{D}[p(y^{(d)}|\mathbf{f}^{(d)}, \sigma^2)p(\mathbf{f}^{(d)}|\mathbf{g}, Z)]p(\mathbf{g}|Z)\prod_{k=1}^{K}[\prod_{i=1}^{N} p(z_{ik}|\theta_k, Q(\cdot, y_i))p(\theta_k|\alpha_k)p(\alpha_k)]
\end{aligned}
\tag{13}
$$

We propose to adopt the Rao-Blackwellized sampling scheme through analytic marginalization from the joint distribution of $\{\mathbf{f}^{(1:D)}\}$ and $\mathbf{g}$, and sample them from their respective posteriors. This improves the efficiency of our Gibbs sampler by reducing the underlying sample space and the variance of later estimates. The conjugate priors result in closed-form marginalization.

By Combining the likelihood marginalized over $\mathbf{f}^{(d)}$ in (11) and the prior in (5), we sample $\mathbf{g}$ from its posterior as

$$
\begin{aligned}
p(\mathbf{g}|Y, Z) &\propto N(\mathbf{g}|\mu_{g|y}, \Sigma_{g|y}) \\
\Sigma_{g|y}^{-1} &= \Sigma_g^{-1} + Z^T \Sigma_{y|g}^{-1} Z \qquad \mu_{g|y} = \Sigma_{g|y} Z^T \Sigma_{y|g}^{-1} \bar{y}
\end{aligned}
\tag{14}
$$

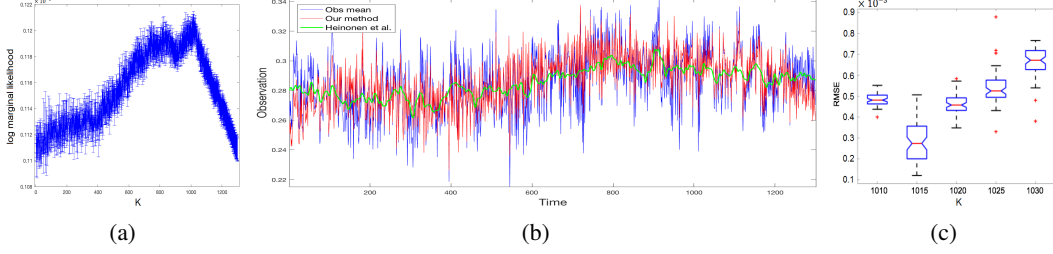

(a)　　　　　　　　　　　　(b)　　　　　　　　　　　　(c)

Figure 1: (a) Plot of mean and $\pm 1$ std of the log marginal likelihood in (12) of the true positive iEEG observations in the training set versus different $K$. (b) Empirical mean of the 8 PIB features of a true positive iEEG observation's heldout segment (blue), our method's predictive mean of the corresponding $y_*^{(1:D)}$ in (24) (red), and the predictive mean of Heinonen et al. method [16] (green). (c) Boxplots of the cross-validation RMSEs summarizing the true positive observations in the training set versus $K$.

By marginalizing over $\mathbf{g}$, each $\mathbf{f}^{(d)}$ has the following posterior distribution:

$$p(\mathbf{f}^{(d)}|y^{(d)}, \mathbf{f}^{(-d)}, Z, \sigma^2) \propto p(y^{(d)}|\mathbf{f}^{(d)}, \sigma^2)p(\mathbf{f}^{(d)}|\mathbf{f}^{(-d)}, Z) \tag{15}$$

where $\mathbf{f}^{(-d)}$ denote the set $\{\mathbf{f}^{(1:D)}\}$ other than $\mathbf{f}^{(d)}$. The first term $p(y^{(d)}|\mathbf{f}^{(d)}, \sigma^2)$ is as in (9), and for the second term, we have

$$p(\mathbf{f}^{(d)}|\mathbf{f}^{(-d)}, Z) = \int p(\mathbf{f}^{(d)}|\mathbf{g}, Z)p(\mathbf{g}|\mathbf{f}^{(-d)}, Z)d\mathbf{g} \tag{16}$$

Recalling (10) and (5), the conditional distribution of $\mathbf{g}$ in (16) is

$$p(\mathbf{g}|\mathbf{f}^{(-d)}, Z) \propto p(\mathbf{f}^{(-d)}|\mathbf{g}, Z)p(\mathbf{g}|Z) = N(\mathbf{g}|\mu_{g|f^{(-d)}}, \Sigma_{g|f^{(-d)}})$$

$$\Sigma_{g|f^{(-d)}}^{-1} = \Sigma_g^{-1} + Z^T(\frac{\Sigma_f}{D-1})^{-1}Z \qquad \mu_{g|f^{(-d)}} = \Sigma_{g|f^{(-d)}}Z^T(\frac{\Sigma_f}{D-1})^{-1}\bar{\mathbf{f}}^{(-d)} \tag{17}$$

where $\bar{\mathbf{f}}^{(-d)} = \frac{\sum_{d'\neq d}\mathbf{f}^{(d')}}{D-1}$. Thus, we have the conditional distribution of $\mathbf{f}^{(d)}$ as

$$\begin{aligned} p(\mathbf{f}^{(d)}|\mathbf{f}^{(-d)}, Z) &= \int p(\mathbf{f}^{(d)}|\mathbf{g}, Z)p(\mathbf{g}|\mathbf{f}^{(-d)}, Z)d\mathbf{g} \\ &= N(\mathbf{f}^{(d)}|Z\mu_{g|f^{(-d)}}, \Sigma_f + Z\Sigma_{g|f^{(-d)}}Z^T) \end{aligned} \tag{18}$$

and the posterior distribution of $\mathbf{f}^{(d)}$ as

$$\begin{aligned} p(\mathbf{f}^{(d)}|y^{(d)}, \mathbf{f}^{(-d)}, Z, \sigma^2) &= N(\mathbf{f}^{(d)}|\mu_{f^{(d)}|f^{(-d)}}, \Sigma_{f^{(d)}|f^{(-d)}}) \\ \Sigma_{f^{(d)}|f^{(-d)}}^{-1} &= (\Sigma_f + Z\Sigma_{g|f^{(-d)}}Z^T)^{-1} + (\sigma^2 I)^{-1} \\ \mu_{f^{(d)}|f^{(-d)}} &= \Sigma_{f^{(d)}|f^{(-d)}}[(\sigma^2 I)^{-1}y^{(d)} + (\Sigma_f + Z\Sigma_{g|f^{(-d)}}Z^T)^{-1}Z\mu_{g|f^{(-d)}}] \end{aligned} \tag{19}$$

We marginalize over $\{\mathbf{f}^{(1:D)}\}$ to sample $z_{ik}$ from its posterior by combining the marginal likelihood in (11) and the prior in (1):

$$\begin{aligned} p(z_{ik} = 1|Y, Z_{-ik}, \mathbf{g}, \sigma^2, \{\theta_k\}, Q) &\propto p(Y|\mathbf{g}, Z, \sigma^2)\prod_i\prod_k p(z_{ik} = 1|\theta_k, Q(\cdot, y_i)) \\ &\propto \prod_d N(y^{(d)}|Z\mathbf{g}, \Sigma_{y|g})\prod_i\prod_k \exp(\theta_k^T Q(\cdot, y_i)) \end{aligned} \tag{20}$$

where $Z_{-ik}$ denotes the matrix $Z$ other than element $z_{ik}$. For binary random variables, Metropolis-Hastings (MH) algorithm is shown to mix faster and have greater statistical efficiency than standard Gibbs samplers [20]. To update $z_{ik}$ given $Z_{-ik}$, we thus use the posterior of (20) to evaluate a MH proposal which flips the binary variable $z_{ik}$ with the current value $z$ to its complement value $\bar{z}$:

$$z_{ik} \propto \kappa(\bar{z}|z)\delta(z_{ik}, \bar{z}) + (1 - \kappa(\bar{z}|z))\delta(z_{ik}, z)$$

$$\kappa(\bar{z}|z) = \min\{\frac{p(z_{ik} = \bar{z}|Y, Z_{-ik}, \sigma^2, \{\theta_k\}, Q)}{p(z_{ik} = z|Y, Z_{-ik}, \sigma^2, \{\theta_k\}, Q)}, 1\} \tag{21}$$

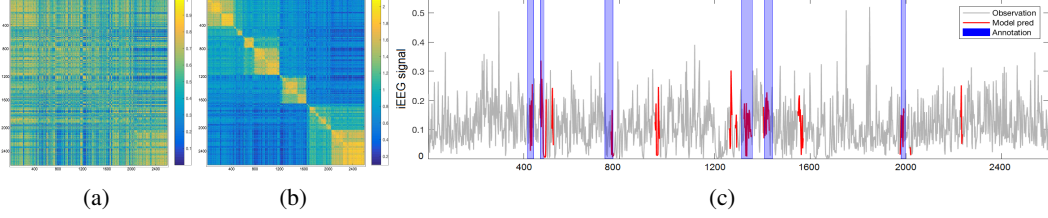

<div align="center">(a)        (b)        (c)</div>

Figure 2: (a) Absolute correlation matrix of a heldout iEEG observation with SOZ events. (b) The corresponding posterior covariance matrix of $\mathbf{f}^{(1:D)}$ with the diagonal blocks as the local covariance matrices of the observation pieces averaged over the Gibbs samples. (c) The blue bands indicate the epileptologist's labels on the SOZ events of the iEEG signal (gray), and the red segments are the encoded pieces predicted to be SOZ events.

To compute the conditional posterior of a coefficient vector $\theta_k$, we fix the set $\{\theta_{-k}\}$ other than $\theta_k$ and have

$$p(\theta_k|Z, \{\theta_{-k}\}, \alpha_k, Q) \propto \prod_i p(z_{ik}|\{\theta_k\}, Q)p(\theta_k|\alpha_k) \propto N(\theta_k|0, A_k^{-1}) \prod_i \eta_{ik}^{\delta(z_i=k)} (1 - \eta_{ik})^{\delta(z_i \neq k)} \tag{22}$$

where $\eta_{ik} \propto \exp[\theta_k^T Q(\cdot, y_i) - \log \sum_{k' \neq k} \exp(\theta_{k'}^T Q(\cdot, y_i))]$. We adopt the logistic sampling technique with auxiliary variable sampling for its efficiency [21].

Finally, given $\{\theta_k\}$ and recalling that each $\alpha_k$ is gamma distributed, its posterior is

$$p(\alpha_k|\theta_k, a, b) = \Gamma(a + \frac{|S_k|}{2}, b + \frac{\sum_{i,k} \theta_{ik}^2}{2}) \tag{23}$$

The set $S_k$ contains the indices for which $\theta_{ik}$ has prior precision $\alpha_k$.

From (11) and (14), the predictive distribution of new observations $y_*^{(d)}$ for the $d$'th variable is

$$p(y_*^{(d)}|Y, Z) = \int p(y_*^{(d)}|\mathbf{g}, Z, \sigma^2)p(\mathbf{g}|Y, Z)d\mathbf{g} = N(y_*^{(d)}|Z\mu_{g|y}, Z\Sigma_{g|y}Z^T + \Sigma_{y|g}) \tag{24}$$

The computational complexity for the predictive is $O(M^2 N)$ due to the block structure of the covariance matrix. After precomputation, the per-iteration complexity is reduced to $O(M^2)$.

## 4 Experiments

We test our method across two domains. For seizure onset localization, we leverage our model to detect early seizure discharges characterized by irregular covariance changes in iEEG recordings. For crime occurrence prediction, our model captures the sharp transitions in regional crime occurrence covariances.

### 4.1 iEEG data description

The dataset of iEEG recordings for SOZ detection are from 83 epilepsy patients [2]. The patients with different SOZs are surgically implanted with different numbers of iEEG sensors in potentially epileptogenic regions in the brains. Among 4966 electrodes in total, 911 of them identified to be in SOZs by clinical epileptologists are taken as true positive examples. The iEEG data are down-sampled to 5 kHz, and filtered to remove artifacts. We adopt power-in-band (PIB) features measuring iEEG data's spectral power in the 8 frequency bands: Delta (0-3Hz), low-theta (3-6 Hz), high-theta (6-9 Hz), alpha (9-14 Hz), beta (14-25 Hz), low-gamma (30-55 Hz), high-gamma (65-115 Hz), and ripple (125-150 Hz), as in [3]. The PIB features extracted from every second in a 2-hour iEEG recording construct an observation $Y$ with $D = 8$ feature variables and length of $N = 7200$.

### 4.2 MCMC settings

---

<sup>2</sup>The dataset is available in `ftp://msel.mayo.edu/EEG_Data/`

<div align="center">6</div>

For each observation, we simulate 3 chains of 7000 Gibbs iterations, and discard the first 3000 as burn-in phase. Each sampling chain is initialized with parameters sampled from their priors. We set $\Gamma(a,b)$ prior on the ARD precisions as $a = |S_k|$ and $b = a/1000$, where $S_k$ is defined in (23). This prior specification is equally informative for various choices of effective coefficient number $|S_k|$ by fixing the prior mean of the prior distribution. Given the number of pieces $K$ fixed, the marginal likelihood in (12) is a function of the hyperparameters $\{l_g, \sigma_g^2, \sigma_f^2\}$. We use empirical Bayes approach to determine the optimum hyperparameter values by optimizing the log marginal likelihood [3]. To determine $K$, we evaluate the marginal likelihood of the true positive iEEG observations in the training set as shown in Figure 1 (a). It suggests $K \approx 1010$ is

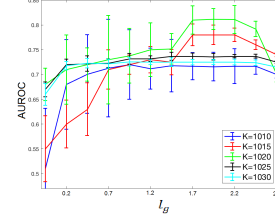

Figure 3: The mean and $\pm 1$ std of AUROC scores for different lengthscale $l_g$ and $K$.

sufficient to capture the covariance variability. We perform the Gelman-Rubin diagnostic [22] to assess convergence by calculating the within-chain and between-chain variances on the Gibbs samples of the posteriors.

## 4.3 SOZ localization

We evaluate SOZ localization as a binary classification task in terms of SOZ abnormal events predicted to be present or absent in an iEEG channel's observation, and use standard performance metrics to compare with the state-of-the-art methods in Table 1.

We use 10-fold cross-validation (CV) to evaluate predictions with 30% test set while keeping the same proportion of SOZ and non-SOZ observations in both sets. We first evaluate our model's regression performance as demonstrated in Figure 1 (b) and (c). Figure 1 (b) shows that both the long-range trend and the changes in covariance smoothness are captured without over-smoothing the local GPs. We summarize the regression performance to fine-tune $K$ in Figure 1 (c) based on the discussion in Section 4.2. Our method encodes each observation's correlations into a covariance matrix with diagonal block structures, as illustrated in Figure 2 (a)-(b). In Figure 2 (c) the pieces capturing SOZ abnormal events are identified through a clinical epileptologist's visual inspection of the true positive iEEG signals. We utilize the local posterior covariance matrices, illustrated in Figure 2 (b), associated with SOZ events as the features of the positive examples, and the local covariances without SOZ events as the negative ones. The averaged similarities of the local covariances in the test set to the positive and negative examples are calculated via the Wasserstein metric: $\|\mu_f - \mu_{f'}\|_2^2 + tr(\Sigma_f + \Sigma_{f'} - 2(\Sigma_{f'}^{\frac{1}{2}} \Sigma_f \Sigma_{f'}^{\frac{1}{2}})^{\frac{1}{2}})$, respectively, and the ratios are used to predict whether an observation consists of SOZ related pieces.

Table 1: Performance evaluation of the SOZ channel detection

| Methods | AUROC | Precision | Recall (Sensitivity) | F1 score |
|---|---|---|---|---|
| **Our method** | $0.80 \pm 0.05$ | $0.41 \pm 0.07$ | $0.75 \pm 0.06$ | $0.51 \pm 0.07$ |
| Factor graph model [3] | $0.72 \pm 0.03$ | $0.39 \pm 0.05$ | $0.74 \pm 0.03$ | $0.46 \pm 0.04$ |
| HFO biomarker [12] | $0.66 \pm 0.07$ | $0.34 \pm 0.05$ | $0.53 \pm 0.08$ | $0.41 \pm 0.04$ |
| Partition-based nonstationary models | | | | |
| N-cuts based mGP [9] | $0.65 \pm 0.03$ | $0.61 \pm 0.02$ | $0.35 \pm 0.07$ | $0.43 \pm 0.05$ |
| Tree based method [6] | $0.64 \pm 0.08$ | $0.59 \pm 0.03$ | $0.38 \pm 0.05$ | $0.40 \pm 0.03$ |
| Nonstationary covariance function models | | | | |
| Paciorek et al. [17] | $0.63 \pm 0.07$ | $0.41 \pm 0.05$ | $0.43 \pm 0.09$ | $0.39 \pm 0.04$ |
| Heinonen et al. [16] | $0.67 \pm 0.05$ | $0.43 \pm 0.03$ | $0.58 \pm 0.03$ | $0.42 \pm 0.07$ |

Figure 3 further explores the interactions between $K$ and $l_g$ around its optimum in terms of the classification performances. The $K$ leading to the best performance is consistent with the regression performance in Figure 1 (c). In Table 1, the factor graphical model method heuristically divides the iEEG recordings into non-overlapping three-second epochs to accommodate SOZ events [3], whereas our method is more flexible by learning the SOZ pieces with various lengths. We implement the other partition-based methods with the same settings as ours. Since these methods can only model

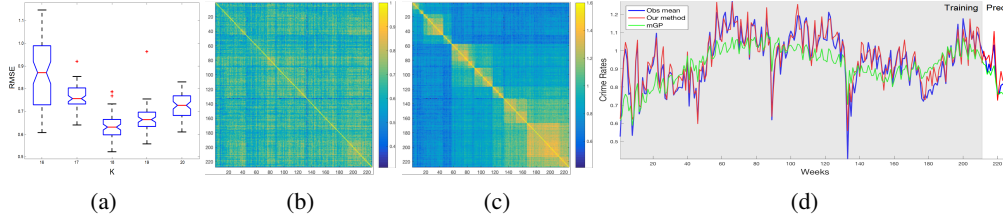

|   (a)   |   (b)   |   (c)   |   (d)   |

Figure 4: (a) RMSE of prediction performance to fine-tune $K$. (b) Absolute correlation matrix of the crime occurrence rates of 179 CTs in 2015-2019. (c) The corresponding posterior covariance matrix of $\mathbf{f}^{(1:D)}$ averaged over the Gibbs samples. (d) plots of observation mean (blue), our method's posterior and predictive mean (red), and N-cuts based mGP [9]'s mean (green).

univariate observations, we apply them on each PIB feature and take the average. For Heinonen et al. method [16], we run 3 chains of 5000 samples of HMC-NUTS sampling to infer the three sets of hyperparameters (noise variance, signal variance, and lengthscale), and initialize the method as suggested. One key to the method is the balance between the signal variance and the nonstationary lengthscale, which is intrinsically related to the partition-based idea. For Paciorek et al. method [17], we use the Matern covariance function described in the paper. The Matern kernel leads to less smooth functions, but it still assumes the covariance structure is the same throughout the entire input space.

## 4.4 Crime event prediction

We apply our method to model the nonstationary evolution of crime occurrence rates in the 179 census tracts (CTs) in Washington, D.C. between 2015-2019 for crime occurrence prediction [4]. We analyze the crime rates on a weekly basis, with totally 227 weeks. By denoting the crime rates in a CT with a variable, we have the multivariate observation $Y$ with $D = 179$ and $N = 227$.

We follow the model setting strategy as in Section 4.2. In particular, We find $K = 15$ with $l_g = 0.5$ sufficiently to account for the crime rates' nonstationary variations based on the predictive performance, as shown in Figure 4 (a). The results in Figure 4 (b)-(d) indicate that we are able to capture the abrupt changes in covariance structure of the CTs' crime rates over time via the posterior and the predictive estimates of $y_*^{(1:D)}$. Figure 4) (d) shows that the classic nonstationary method mGP [9] tends to over-smooth the local covariance variability for combining a global GP with the local GPs.

We predict the one-week-ahead crime rates in each tract for the first 16 weeks in 2019 based on the posterior estimates in 2015-2018. We estimate the posterior predictive of the 2019 weekly crime rate in each CT $y_*^{(d)}$ as in (24) by averaging over the Gibbs samples. Table 2 shows the monthly-averaged prediction RMSEs, conditioned on the observations in 2015-2018. For PoINAR, we use the same setting as in [15]. The implementation of Paciorek et al. method [17] and Heinonen et al. method [16] are the same as in Section 4.3. One major challenge to implement Paciorek et al. method is that the number of its hyperparameters increases fast in multivariate cases. In particular, computation of the kernel matrices at each input location is slow because of the matrix decomposition ($O(D^3)$). In contrast, our method is more computationally efficient by introducing the conditional independence given the hyper-GP as in (10). The results indicate that our method produces lower RMSE. Figure 5 visualizes the RMSEs between our method's predictions and the ground truth by CTs geographically.

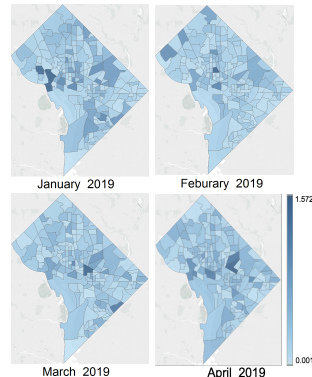

Figure 5: 2019 monthly averaged RMSE maps between the ground truth and our model's predictive means of $y_*^{(1:D)}$.

Table 2: Monthly average RMSE of one-week-ahead predictions of the crime rates in 2019.

| $RMSE \pm error$ | Jan. 2019 | Feb. 2019 | Mar. 2019 | April 2019 |
|---|---|---|---|---|
| **Our method** | $0.638 \pm 0.025$ | $0.707 \pm 0.023$ | $0.815 \pm 0.029$ | $0.817 \pm 0.027$ |
| N-cuts based mGP [9] | $0.657 \pm 0.023$ | $0.818 \pm 0.019$ | $0.893 \pm 0.033$ | $1.071 \pm 0.032$ |
| PoINAR [15] | $0.839 \pm 0.017$ | $0.825 \pm 0.014$ | $0.912 \pm 0.086$ | $1.165 \pm 0.006$ |
| Paciorek et al. [17] | $0.949 \pm 0.034$ | $1.122 \pm 0.055$ | $1.176 \pm 0.209$ | $1.462 \pm 0.147$ |
| Heinonen et al. [16] | $0.704 \pm 0.031$ | $0.875 \pm 0.118$ | $0.931 \pm 0.763$ | $1.069 \pm 0.014$ |

## 5  Conclusions

Our unified nonstationary modeling framework integrates a sparse encoding process that transforms the observations into a piece-wise representation with a hyper GP defined over its relevance vectors. The hyper GP governs a set of local GPs fitted to the pieces through their mean functions. The framework efficiently extends to multivariate observations by inducing conditional independence among variables and between their respective local GPs. It achieves superior performance over the state-of-the-art competitors by effectively capturing both sharp changes in covariance smoothness and long-range trend.

## 6  Acknowledgments

This work is funded in part by National Science Foundation (NSF-1850492).

## Footnotes

[1]See the supplementary material for the derivation of 12 and its gradients.

[3]See the supplementary material for the implementation.

[4]The crime data are available on http://opendata.dc.gov

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
