[Supplementary Material · Supplement_neurips_2019.pdf]

# Supplement

**Affiliation**
**Address**
email

## 1 Marginal likelihood derivation

To extend to multivariate observations $Y = \{y^{(1)}, \cdots, y^{(D)}\}$ where $y^{(d)}$ denotes the observations of the $d$th variable (e.g., a feature dimension of EEG recording, a census tract's crime occurrence), By exploiting the conditional independence of $Y$, the marginal likelihood for the multivariate observations is:

$$
\begin{aligned}
p(Y|Z,\sigma^2) &= \int \prod_{d=1}^{D} p(y^{(d)}|\mathbf{g}, Z, \sigma^2) p(\mathbf{g}|C) d\mathbf{g} \\
&= \int \prod^{d} N(y^{(d)}|Z\mathbf{g}, \Sigma_{y|g}) N(\mathbf{g}|0, \Sigma_g) d\mathbf{g} \\
&= \int N(\mathbf{g}|0, \Sigma_g) \prod_{d} (|2\pi\Sigma_{y|g}|^{-\frac{1}{2}} \exp[-\frac{1}{2}(y^{(d)} - Z\mathbf{g})^T \Sigma_{y|g}^{-1}(y^{(j)} - Z\mathbf{g})]) d\mathbf{g} \\
&= \int N(\mathbf{g}|0, \Sigma_g) |2\pi\Sigma_{y|g}|^{-\frac{D}{2}} \exp[-\frac{1}{2}\sum_{d}(y^{(d)} - Z\mathbf{g})^T \Sigma_{y|g}^{-1}(y^{(d)} - Z\mathbf{g})] d\mathbf{g} \\
&= \int N(\mathbf{g}|0, \Sigma_g) |2\pi\Sigma_{y|g}|^{-\frac{D}{2}} \exp[-\frac{1}{2}(Z\mathbf{g} - \bar{y})^T (\frac{\Sigma_{y|g}}{D})^{-1}(Z\mathbf{g} - \bar{y}) - \frac{1}{2}\sum_{d}(y^{(d)} - \bar{y})^T \Sigma_{y|g}^{-1}(y^{(d)} - \bar{y})] d\mathbf{g} \\
&= \int N(\mathbf{g}|0, \Sigma_g) |2\pi\Sigma_{y|g}|^{-\frac{D}{2}} \exp[-\frac{1}{2}(Z\mathbf{g} - \bar{y})^T (\frac{\Sigma_{y|g}}{D})^{-1}(Z\mathbf{g} - \bar{y})] \exp[-\frac{1}{2}\sum_{d}(y^{(d)} - \bar{y})^T \Sigma_{y|g}^{-1}(y^{(j)} - \bar{y})] d\mathbf{g} \\
&= \int N(\mathbf{g}|0, \Sigma_g) |2\pi\Sigma_{y|g}|^{-\frac{J}{2}} |2\pi\frac{\Sigma_{y|g}}{J}|^{\frac{1}{2}} N(Z\mathbf{g}; \bar{y}, \frac{\Sigma_{y|g}}{D}) \exp[-\frac{1}{2}\sum_{d}(y^{(d)} - \bar{y})^T \Sigma_{y|g}^{-1}(y^{(d)} - \bar{y})] d\mathbf{g} \\
&= \int N(\mathbf{g}|0, \Sigma_g) \frac{|2\pi\frac{\Sigma_{y|g}}{D}|^{\frac{1}{2}}}{|2\pi\Sigma_{y|g}|^{\frac{D}{2}}} N(Z\mathbf{g}; \bar{y}, \frac{\Sigma_{y|g}}{D}) \exp[-\frac{1}{2}tr(\sum_{d}(y^{(d)} - \bar{y})^T \Sigma_{y|g}^{-1}(y^{(d)} - \bar{y}))] d\mathbf{g} \\
&= \frac{|2\pi\frac{\Sigma_{y|g}}{D}|^{\frac{1}{2}}}{|2\pi\Sigma_{y|g}|^{\frac{D}{2}}} \exp[-\frac{1}{2}tr(\sum_{d}(y^{(d)} - \bar{y})(y^{(d)} - \bar{y})^T)\Sigma_{y|g}^{-1}] \int N(\bar{y}; Z\mathbf{g}, \frac{\Sigma_{y|g}}{N}) N(\mathbf{g}; 0, \Sigma_g) d\mathbf{g} \\
&= \frac{|2\pi\frac{\Sigma_{y|g}}{D}|^{\frac{1}{2}}}{|2\pi\Sigma_{y|g}|^{\frac{D}{2}}} \exp[-\frac{1}{2}tr(\sum_{d}(y^{(d)} - \bar{y})(y^{(d)} - \bar{y})^T)\Sigma_{y|g}^{-1}] N(\bar{y}; 0, \frac{\Sigma_{y|g}}{D} + Z\Sigma_g Z^T)
\end{aligned}
\tag{1}
$$

where $\bar{y} = \frac{\sum_d y^{(d)}}{D}$.

## 2 Efficient inference

The posterior distribution of $\mathbf{f}^{(d)}$ as

$$
\begin{aligned}
&p(\mathbf{f}^{(d)}|y^{(d)}, \mathbf{f}^{(-d)}, Z, \sigma^2) \\
&\propto p(y^{(d)}|\mathbf{f}^{(d)}, \sigma^2)p(\mathbf{f}^{(d)}|\mathbf{f}^{(-d)}, Z) \\
&= N(y^{(d)}|\mathbf{f}^{(d)}, \sigma^2 I)N(\mathbf{f}^{(d)}|Z\mu_{g|f^{(-d)}}, \Sigma_f + Z\Sigma_{g|f^{(-d)}}Z^T) \\
&= N(\mathbf{f}^{(d)}|\mu_{f^{(d)}|f^{(-d)}}, \Sigma_{f^{(d)}|f^{(-d)}}) \\
\Sigma_{f^{(d)}|f^{(-d)}}^{-1} &= (\Sigma_f + Z\Sigma_{g|f^{(-d)}}Z^T)^{-1} + (\sigma^2 I)^{-1} \\
\mu_{f^{(d)}|f^{(-d)}} &= \Sigma_{f^{(d)}|f^{(-d)}}[(\sigma^2 I)^{-1}y^{(d)} + (\Sigma_f + Z\Sigma_{g|f^{(-d)}}Z^T)^{-1}Z\mu_{g|f^{(-d)}}]
\end{aligned}
\tag{2}
$$

## 3 Estimating the kernel parameters

To estimate the hyper-parameters of the kernels, we adopt the empirical Bayes approach for continuous optimization methods. In particular, we maximize the marginal likelihood by marginalizing out the latent function $f$. This moves us up one level of the Bayesian hierarchy, and reduces the chances of overfitting, since the number of kernel parameters is fairly small in our setting. $Z$ specifies the covariance structure in terms of the combinations of $\kappa_g$ and $\kappa_f$. Both of the kernel functions are parametrized by the hyperparameters $\{l_g, \sigma_g^2, \sigma_f^2\}$. We thus illustrate our method on a more general form of $p(Y|Z, \sigma^2)$:

The exponential term of Eqn. (1) $-\frac{1}{2}\bar{y}^T(\frac{\Sigma_{y|g}}{D} + Z\Sigma_g Z^T)\bar{y}$ measures how well the model fits the data, the normalizing constant term $\log|2\pi\frac{\Sigma_{y|g}}{D}|^{\frac{1}{2}}| - \log|2\pi\Sigma_{y|g}|^{\frac{D}{2}}$ measures model complexity, and the third one is just a constant. The tradeoff between the first two terms provides us a simpler model fitting the data well. In 1d case of a SE kernel, as the length scale $\ell$ varies and hold $\sigma_y^2$ fixed. For short length scales, the fit will be good, so $-\frac{1}{2}\bar{y}^T(\frac{\Sigma_{y|g}}{D} + Z\Sigma_g Z^T)\bar{y}$ will be small. However, the model complexity will be high: $K$ will be almost diagonal and most points will not be considered "near" any others, so the $\log|2\pi\frac{\Sigma_{y|g}}{D}|^{\frac{1}{2}}| - \log|2\pi\Sigma_{y|g}|^{\frac{D}{2}}$ will be large. For long length scales, the fit will be poor but the model complexity will be low: $K$ will be almost all $1's$, so $\log|2\pi\frac{\Sigma_{y|g}}{D}|^{\frac{1}{2}}| - \log|2\pi\Sigma_{y|g}|^{\frac{D}{2}}$ will be small.

To maximize the marginal likelihood, let the hyper-parameters be denoted by $\theta = \{l_g, \sigma_g^2, \sigma_f^2\}$. By defining $K_y = \frac{\Sigma_{y|g}}{D} + Z\Sigma_g Z^T$, we can show that

$$
\begin{aligned}
\frac{\partial}{\partial\theta_j}\log p(y|X) &= \frac{1}{2}y^T K_y^{-1}\frac{\partial K_y}{\partial\theta_j}K_y^{-1}y - \frac{1}{2}tr(K_y^{-1}\frac{\partial K_y}{\partial\theta_j}) \\
&= \frac{1}{2}tr(((K_y^{-1}y)(K_y^{-1}y)^T - K_y^{-1})\frac{\partial K_y}{\partial\theta_j})
\end{aligned}
\tag{3}
$$

This derivation needs the rules: $\frac{\partial}{\partial x}A(x)^{-1} = -A(x)^{-1}(\frac{\partial}{\partial x}A(x))A(x)^{-1}$ and $\frac{\partial}{\partial x}|A(x)| = |A(x)|tr(A(x)^{-1}\frac{\partial}{\partial x}A(x))$. It takes $O(N^3)$ time to compute $K_y^{-1}$, and then $O(N^2)$ time per hyperparameter to compute the gradient. To meet the constraints on the hyper-parameters, such as $\sigma_y^2 \geq 0$, so we define $\theta = \log(\sigma_y^2)$, and then use the chain rule. Given an expression for the log marginal likelihood and its derivative, we can estimate the kernel parameters using any standard gradient-based optimizer. However, since the objective is not convex, local minima can be a problem. Alternatively, we can adopt Hamiltonian Monte Carlo method to sample the hyperparameters.

## 4 More experiment results

(a) log marginal likelihood Vs. K     (b) observation PIB features     (c) prediction PIB features

Figure 1: (a) Log marginal likelihoods of the negative iEEG observations without SOZ events with different K. (b) Spectral characteristics of a patient's positive iEEG observation's PIB features with SOZ events (top) and negative iEEG observation's PIB features without SOZ events (bottom). (c) Spectral characteristics of our model's predictions on the corresponding positive and negative observations averaged over 4000-7000 random samples of 3 Gibbs sampling chains.

(a) crime rates        (b) posterior means & obs. mean

Figure 2: (a) Crime occurrence rates of the 179 CTs in Washington D.C. in 2015-2019. (b) Posterior means of the 179 CTs over time and the empirical mean (red).

(a) Jan. 2019      (b) Feb. 2019      (c) Mar. 2019      (d) April 2019

Figure 3: Monthly averaged crime rates maps of the ground truth (left) and the corresponding maps of predictive posterior mean rates $y_*^{(1:D)}$ using the samples from 4000 to 7000 iterations of the 3 Gibbs sampling chains (right) in 2019.