[Reviews · NeurIPS 2019]

Reviewer 1



The authors provide a model of stochastic processes that aim at modeling strong non-stationarities. The model is built from Gaussian Processes (GPs) and has multivariate outputs. Algorithms are carefully discussed for the Bayesian implementation. Good results are provided on two real applications to iEEG and criminology. Originality: As discussed in the paper, the model is more flexible than previous approaches for modeling non-stationarity. The method and applications of the paper are thus original, in my opinion. Quality: The model is complex and seems to provide good numerical results in applications. The methodology is well developed. Clarity: I think the clarity could be improved. I would have liked to see more careful definitions of the different quantity involved, and more explanations and discussions. For instance around equation (3) I did not understand what C is exactly. Significance: I think the topic is significant and the methods of the paper are useful.

Reviewer 2



Originality: As mentioned in the paper, when dealing with non-stationary data within GPs, there are two approaches: 1. Use a non-stationary covariance function and 2. Partition the input space into local regions and for each region fit a stationary GP. The paper follows the latter, although the novelty of the paper is the fact that the probabilistic partition is coupled with ARD which yields the most relevant data point for each partition member, which become the training locations for the global GP. The paper lack comparisons against the first type of non-stationary GP models. Clarity: I found the presentation of the paper very clear and easy to follow. It would have been ideal to see, for the sake of comprehension, an example showing the partitions learned with each local GP on it and the long-range global GP. Quality: The paper is technically correct although there are some potential issues that limit the effectiveness of the proposed model. For instance, the multivariate model is limited to highly correlated signals since the underlying global GP is the same across all multivariate variables. A mixture of global GPs could allow for more flexibility across signals, as is typically done in multivariate GPs models. Questions: - Given that the probabilistic model for the partitions is based on observation correlations, how does the model guarantee that each partition is local? - Do you think is possible to encode the observation correlations into a Chinese Restaurant process in order to determine the number of partitions nonparametrically? Significance: I can see this model being used in the problem of SOZ detection given its principled approach and the experimental results, although the scope of the model seems rather limited to applications to highly correlated signals.

Reviewer 3



Generally, the idea is clearly conveyed. Although the techniques employed are not new, the modelling is quite intuitive, and all derivations are technically sound. However, there is a major problem with the rigor of the related work section. The paper seems to have missed an important work on non-stationary GP which imposes hyper GP priors on the parameters of the squared exponential covariance function (Non-Stationary Gaussian Process Regression with Hamiltonian Monte Carlo, Heinonen et al., 2016). I believe this work does not belong in the same category with [14] (which the authors have claimed to be “too strong of a modeling assumption”), because it jointly learns the distribution of the covariance parameters. I feel that the referred work is relevant and deserves to be rigorously compared with the proposed method in the experiment section. As for [14], the authors have also not justified their claim with empirical evidence. Most comparisons are made against stationary methods such as full GP and local stationary GP ensemble with a separate partitioning step. I would like to see how [14] proves to be ineffective in the domain of interest because of its restrictive assumption. It would also be helpful if the author can show RMSE vs. no. Gibbs sampling iterations to demonstrate the convergence of the proposed method. -- Post-rebuttal Feedback: Thank you for the response, which has reasonably addressed my concern. I have increased my rating of your work.

[Author Response · NeurIPS 2019]

We'd like to thank all the reviewers for your time and your constructive comments. They are valuable for our future
research.

we thank Reviewer 1 (**R1**) for pointing out the clarity of notation definitions. We will provide more explanations for
the quantity definitions in the camera-ready version. As in p.98-99, $C$ denotes a subset of input $X$, whose elements'
corresponding observations are the relevance vectors (derived from ARD in Eq. (2)) with the maximum non-zero
component of the weight vector $\theta_k$.

As **R2** and **R3** suggested, we implement two nonstationary covariance function methods in the two applications for
empirical comparisons. One is a classical nonstationary Gaussian process regression: Paciorek et al. "Nonstationary
covariance functions for Gaussian process regression", which is cited as [14] in our paper. The other is Heinonen et al.
"NonStationary Gaussian Process Regression with Hamiltonian Monte Carlo" coupling nonstationary GP with priors
on the hyperparameters of the squared exponential covariance functions. We implement these methods with the same
experimental settings as ours. In particular, since Heinonen et al. can only model univariate observations, we have to
apply it to model each observation variable independently and evaluate the means for the multivariate cases.

Table 1: Performance evaluation of the SOZ channel detection

| Methods | AUROC | Precision | Recall (Sensitivity) | F1 score |
|---|---|---|---|---|
| **Our method** | $0.81 \pm 0.05$ | $0.45 \pm 0.07$ | $0.77 \pm 0.06$ | $0.51 \pm 0.07$ |
| Nonstationary covariance function models | | | | |
| Paciorek et al. | $0.63 \pm 0.07$ | $0.41 \pm 0.05$ | $0.43 \pm 0.09$ | $0.39 \pm 0.04$ |
| Heinonen et al. | $0.67 \pm 0.05$ | $0.43 \pm 0.03$ | $0.58 \pm 0.03$ | $0.42 \pm 0.07$ |

Table 2: Monthly average RMSE of one-week-ahead predictions of the crime rates in 2019.

| $RMSE \pm error$ | Jan. 2019 | Feb. 2019 | Mar. 2019 | April 2019 |
|---|---|---|---|---|
| **Our method** | $0.638 \pm 0.025$ | $0.707 \pm 0.023$ | $0.815 \pm 0.029$ | $0.817 \pm 0.027$ |
| Paciorek et al. | $0.949 \pm 0.034$ | $1.122 \pm 0.055$ | $1.176 \pm 0.209$ | $1.462 \pm 0.147$ |
| Heinonen et al. | $0.704 \pm 0.031$ | $0.875 \pm 0.118$ | $0.931 \pm 0.763$ | $1.069 \pm 0.014$ |

(a)      (b)      (c)

Figure 1: (a) Empirical mean of the 8 PIB features of a true positive iEEG observation's heldout segment (blue), the
predictive mean of our method (red), and the predictive mean of Heinonen et al. method (green). (b) Plots of observation
mean (blue) of the crime occurrence rates of 179 CTs in 2015-2019, our method's posterior and predictive mean (red),
and Heinonen et al. mean (green). (c) Plot of RMSE vs. 6 Gibbs sampling chains for the convergence.

As in Tables 1 and 2, both methods are inferior to ours, but Heinonen et al. method is comparable to the partition-based
nonstationary methods. For Heinonen et al. method, we run 3 chains of 5000 samples of HMC-NUTS sampling to
infer the three sets of hyperparameters (noise variance, signal variance, and lengthscale), and initialize the method as
suggested in the paper. As in Figure 1 (a) and (b), one key to the success of the method is the balance between the
signal variance and the nonstationary lengthscale, which is intrinsically related to the partition-based idea. For Paciorek
et al. method, we use the Matern covariance function described in the paper. The Matern kernel leads to less smooth
functions, but it still assumes a stationary process in that the covariance structure is the same throughout the entire input
space. One major challenge to implement Paciorek et al. method is that the number of hyperparameters increases fast
in multivariate cases. In particular, computation of the kernel matrices at each input location is slow because of the
matrix decomposition ($O(n^3)$). In contrast, our method is more computationally efficient by introducing the conditional
independence given the hyper-GP as in Eq. (10). As suggested by **R3**, we plot the RMSE vs. Gibbs samples in Figure 1
(c) to demonstrate the convergence for DC crime case. We'll add these results to the camera-ready version. To **R2**, if
partitions are not local, the method will reduce to a stationary one. Chinese restaurant process will be our future work.

[Meta-Review · NeurIPS 2019]

The GP-based method for modelling non-stationary data is interesting. The combination of a global GP and local GPs by learning sparse multinomial logit/softmax seems to work well in practice. The authors have added more experiments in the rebuttal that addressed some of the reviewers' concerns. I would suggest to discuss further in the related work the connection with Mixture of Gaussian Process Experts (see e.g Rasmussen and Grahramani 2002, Meeds and Osindero).